# Reassessment of the Matrix Composition of Co-Re-Cr-Based Alloys for Particle Strengthening in High-Temperature Applications and Investigation of Suitable MC-Carbides

**DOI:** 10.3390/ma16124443

**Published:** 2023-06-17

**Authors:** Eugen Seif, Joachim Rösler

**Affiliations:** Institut für Werkstoffe, TU Braunschweig, 38106 Braunschweig, Germany; e.seif@tu-braunschweig.de

**Keywords:** high-temperature alloys, Co-Re alloys, TaC, TiC, HfC

## Abstract

This study reassesses the optimum matrix composition of Co-Re-Cr-based alloys for strengthening by MC-type carbides. It is found that the composition of Co-15Re-5Cr is ideally suited for this purpose as it allows the solution of the carbide-forming elements such as Ta, Ti, Hf, and C within a matrix consisting entirely of fcc-phase (typically at 1450 °C), having a high solubility for these elements, while precipitation heat treatment (typically at 900–1100 °C) occurs in a hcp-Co matrix, displaying a much lower solubility. In the case of the monocarbides TiC and HfC, this was investigated and achieved for the first time in Co-Re-based alloys. TaC and TiC emerged as suitable particles in Co-Re-Cr alloys for creep applications due to a large population of nano-sized particle precipitation, which is not the case for the mainly coarse HfC. Both Co-15Re-5Cr-xTa-xC and Co-15Re-5Cr-xTi-xC exhibit a formerly unknown maximum solubility close to x = 1.8 at.%. Therefore, further research on the particle-strengthening effect and the governing creep mechanisms of carbide-strengthened Co-Re-Cr alloys should focus on alloys with the following compositions: Co-15Re-5Cr-1.8Ta-1.8C and Co-15Re-5Cr-1.8Ti-1.8C

## 1. Introduction

Decades of extensive research on Ni-based superalloys have rendered this material class the preferred choice when it comes to high-temperature applications in combustion-based energy production and aircraft transportation. These superalloys combine the benefits of good mechanical properties such as creep strength and fracture toughness with oxidation and corrosion resistance and broad manufacturing knowledge. However, their maximum operation temperature is fundamentally limited by their relatively low solidus temperature of about 1300 °C, giving rise to the need for alternative alloys with the potential for higher application temperatures by virtue of an increased solidus temperature since the energy efficiency of the combustion process heavily depends on the maximization of the combustion gas temperatures [1]. Up until now, high combustion gas temperatures (close to and above the solidus temperatures of Ni-based superalloys) were merely endured by Ni-based superalloys with an additional thermal barrier coating and turbine blade cooling [2]. In recent years, due to environmental issues, general demands for the reduction of carbon emissions have emerged. In terms of the stated fields of industry, this can be achieved by the use of materials that provide at least a comparable strength as Ni-based superalloys but at higher temperatures, so that a further increase in the combustion gas temperatures becomes technologically feasible. Therefore, an alternative approach utilizing Co-Re-based alloys (with melting temperatures exceeding those of Ni-based superalloys by approx. 100 °C) was introduced in the late 2000s, where large quantities of rhenium (Re) (up to 31 at.%) were added to Co [3]. Re dissolved in Co increases the strength due to solid solution strengthening [4], retards diffusion more than other refractory metals such as Mo, Ta, and W [5], and raises the melting temperature [6]. All these properties are necessary for high-temperature applications where dislocation creep and/or diffusional creep govern the high-temperature deformation mechanisms [7]. The complete miscibility of Co and Re provides a further benefit as it allows a wide range of compositions tailored to the specific requirements of the application [6]. Nonetheless, alloys solely strengthened by solid solution hardening simply cannot meet the strength levels required in demanding high-temperature applications. Therefore, additional strengthening by precipitation hardening has to be considered. The first studies on suitable precipitates in the Co-Re-system focused on the formation of Cr_23_C_6_ and TaC. Cr_23_C_6_ precipitates either as blocky carbides ranging above the µm-scale or in plate-like shape on the nm-scale regarding its thickness but µm-scale in its plane extensions. TaC mainly appears in a fine globular or elongated shape with a diameter < 100 nm. Adding larger amounts of Cr to Co-Re leads to the formation of a further phase besides the Cr-carbides. The topologically close packed phase of Cr_2_Re_3_, also known as σ-phase, tends to form blocky particles even larger than Cr_23_C_6_ [3,4]. However, the σ-phase can be refined by discontinuous (cellular) precipitation if proper heat treatment and additional Ni in the composition are provided [8]. Nevertheless, Cr_2_Re_3_ features several drawbacks, making it less suitable for high-temperature applications. Firstly, its pronounced brittleness facilitates crack propagation along the particle matrix interface [9,10]. Secondly, the formation of Cr_2_Re_3_ depletes the local Co-Re matrix in the Re matrix and thereby reduces its contribution to the solid solution’s strengthening and slows down diffusion [8]. Thirdly, upon completion of the discontinuous precipitation, Cr_2_Re_3_ continuously grows up to large spheroids under creep conditions at 1100 °C [10]. Unfortunately, Cr_23_C_6_ is not a suitable particle type for creep applications due to its inferior thermal stability. Depending on the exact composition of the Co-Re alloy and additional elements, the plate-like Cr-carbides dissolve at temperatures starting at 1000 °C [11], and the dissolved Cr favors even more the disadvantageous formation and coarsening of Cr_2_Re_3_ [12]. So far, TaC has shown the most promising application possibility as it precipitates as fine particles < 100 nm [3,4,9,11,13,14] and interacts with dislocations of the Co-Re matrix under creep conditions [9]. A further benefit derives from the thermal stability of TaC, which is reportedly stable at temperatures up to 1200 °C [13] and, in some cases, even up to approx. 1300 °C [11,13]. Other relevant particle properties for creep applications, such as volume fraction, coarsening behavior, and particle size distribution, heavily depend on the exact chemical composition. At given Ta-content, Co-Re-Cr-alloys with under-stochiometric compositions of the monocarbide-forming elements, meaning C-at.%/Ta-at.% < 1 (= y = C/Ta-ratio), contain lower volume fractions of TaC within the matrix compared to stochiometric compositions as simply fewer carbide-forming atoms are available. Higher volume fractions lead to comparatively larger particle sizes since the particles nucleate and grow slightly faster. However, once precipitation is completed, further coarsening is very sluggish, leading to an essentially stable particle size of about 47 nm (y = 1) and 92 nm (y = 0.9) for aging at 900 °C and 1100 °C for 15 h, respectively. Lower C/Ta-ratios also reduce the temperature range of the allotropic hcp ⇄ fcc phase transformation of the Co-Re-Cr matrix due to the fcc-stabilizing effect of free Ta [15]. Taking into consideration that the solubility of Ta drops significantly in hcp-Co compared to fcc-Co [16], the TaC precipitation during cooling from the solution heat treatment temperature starts with the phase transformation of the matrix from fcc to hcp, i.e., at lower temperatures in the case of C/Ta-ratios < 1. Therefore, less time for the nucleation and growth of TaC is available during cooling after solution treatment when the C/Ta-ratio is reduced, which results in a smaller particle size [15]. Vice versa, the addition of Cr as a hcp-stabilizing element leads to the rise of the hcp ⇄ fcc phase transformation temperature range and consequently produces larger particles [16]. Still, C/Ta-ratios less than 1 are not desirable in the design of TaC-strengthened Co-Re-based alloys because of the risk of retained fcc-Co after the solution heat treatment [15]. Several issues arise from the presence of metastable fcc-Co under creep conditions. Firstly, TaC dissolves in fcc-Co and does not contribute to the creep resistance anymore. Secondly, martensitic and/or diffusion-controlled fcc → hcp transformation during service may lead to grain refinement, adversely affecting creep strength [16]. Finally, the fcc matrix is significantly softer than the hcp matrix [8], and creep processes seem to be faster for Co-Re alloys with more metastable fcc-Co.

Altogether, multiple studies have revealed some favorable properties of TaC-strengthened Co-Re-alloys for creep applications. However, contemporary literature lacks a proper assessment of the particle-strengthening potential of TaC in a hcp Co-Re matrix. At this point, the nature of the particle-dislocation interaction of TaC under dislocation creep conditions is completely unknown. Thus, it is unclear which particle bypassing mechanism (climbing and/or detachment) contributes to what extent to the creep resistance. The majority of the creep models based on dislocation climb and/or dislocation detachment depend in one way or another on the following properties: volume fraction, particle size, interparticle distance, coherency, and dislocation line energy relaxation [7]. The quantitative assessment of TaC-strengthened Co-Re alloys by the experimental determination of the aforementioned properties requires a pre-study to set ideal research conditions concerning the microstructure of the TaC-strengthened Co-Re alloys. In this context, ideal research conditions are defined by the following four requirements: (1) Homogenization and solution treatment take place within a Co-Re matrix consisting exclusively of the fcc-phase, so that its better solubility for the MC-forming elements compared to the hcp-phase can be fully exploited. (2) TaC completely dissolves under solution treatment. (3) No retained fcc-Co after solution treatment. (4) TaC-precipitation by aging occurs in a Co-Re matrix consisting entirely of the hcp-phase.

Previous studies have already focused on the determination of the stated requirements. In-situ neutron diffraction measurements suggested that the composition of Co-17Re-5Cr-1.2-Ta-1.082C (all mentioned compositions throughout this work are stated in at. %) has successfully met these conditions. On heating, the temperatures at which the fcc phase starts to form, referred to in the following as the fcc start temperature, and where hcp is fully transformed to fcc, called the fcc finish temperature, are reached at 1260 °C and 1410 °C, respectively. Furthermore, no retained fcc-Co was found after cooling [17]. This temperature range of the hcp to fcc transformation allows TaC to dissolve in fcc-Co during solution treatment (ST) and precipitate it in hcp-Co at temperatures where significant particle growth happens (e.g., 1100 °C [18]). However, during the reproduction of these results on Co-17Re-5Cr-based alloys during this study, the authors encountered strong evidence that ST for Co-17Re-5Cr and its carbide-strengthened variant (Co-17Re-5Cr-xTa-xC as seen in Figure 8) is not in accordance with the stated observations. As will be demonstrated, the resulting microstructure exhibits strong evidence of retained hcp-Co during ST. Consequently, due to the very low solubility of Ta and C in the hcp phase even under ST conditions, large regions remain free of TaC after precipitation heat treatment. Obviously, these relatively soft regions would adversely affect creep resistance. Furthermore, rejection of Ta and C from the hcp phase may oversaturate the fcc phase, thus favoring the formation of coarse TaC during ST. Consequently, some amounts of Ta and C would be lost for precipitation strengthening.

For these reasons, the main purpose of this study is to reassess the composition and/or heat treatment that achieves the four requirements before a proper assessment of the creep performance of carbide-strengthened Co-Re-based alloys can be conducted. This study intends to provide evidence for the presence of hcp-Co during ST for Co-17Re-based alloys and identify a new Re-reduced composition for which a pure fcc matrix solely exists during ST. Another purpose of the study is to determine the maximum solubility of Ta and C in fcc-Co.

Moreover, new MC-carbides, namely TiC and HfC, are incorporated for the first time in Co-Re-based alloys, which were suggested as promising particles in other Co-based alloys [19]. ZrC was not considered as the maximum solubility of Zr in fcc-Co lies at 0.2 at.% [20], so only a low volume fraction of ZrC precipitates and an insufficient contribution to particle strengthening is expected. Both TiC and HfC may exhibit other interfacial properties with the Co-Re matrix than TaC and may result in different coarsening behavior or particle-dislocation interactions. Particularly, the detachment-controlled dislocation creep process mainly relies on the relaxation of the dislocation line energy at the particle-matrix interface [21] and might differ for TaC, TiC, and HfC as these share the same lattice structure but feature different lattice parameters (0.4456 nm, 0.4328 nm, and 0.4633 nm, respectively) [22]. Furthermore, this study sheds light on the maximum solubility and particle size of TiC and HfC in Co-Re-based alloys.

## 2. Materials and Experiments

Different Co-Re-(Cr)-X-C-alloys with X = Ta, Ti, and Hf were cast in a two-step melting process. Firstly, a master alloy of Co-Re or Co-Re-Cr was melted in a EMA vacuum induction furnace (VIF, Hirschhorn, Germany) already containing the majority of the required Co, Re, and/or Cr of the targeted alloy. Subsequently, the carbide forming elements (Ta, Ti, Hf, and C) as well as the remaining small amounts of Re (and Cr) were added to the master alloy and cast by vacuum arc melting (VAM) in the PINK vacuum arc furnace (Wertheim, Germany) into rods with a diameter of 13 mm and a length of 80 mm. The two-step melting process was first introduced in this study and is supposed to reduce the evaporation of Cr and Co, which have significantly larger vapor pressures than Re. If a single-step VAM process is used instead, the melting of the high amount of Re takes the most time and requires several re-melting steps until it is fully dissolved in the melt. However, as the beam hits the solid Re, the surrounding melt is heated up to the point of partial evaporation of Co and Cr. This is not the case with the VIF process, since severe local overheating does not take place. Instead, Re steadily dissolves by convection and diffusion in the melt. Therefore, the less Re that has to be added to the master alloy in the VAM process, the less melting time and re-melting steps are needed. Thus, less evaporation of Co and Cr occurs.

The first and second master alloys were cast at the Institute for Materials Science at the Technical University Braunschweig with a nominal composition of Co-17Re-5Cr and Co-15Re-5Cr, respectively, whereas the third master alloy with a composition of Co-20Re was produced by VDM Metals International GmbH in Werdohl (Germany). After the first cast of Co-Re-Cr-Ta/Ti/Hf-C by VAM using the first master alloy, EDX measurements conducted with a Hitachi TM3000 scanning electron microscope (Tokyo, Japan) revealed that the existent amount of Re was significantly less than the intended 17 at.%. Additional EDX measurements of the remains of the master alloy Co-17Re-5Cr showed that the distribution of Re was inhomogeneous and either surpassed or failed to reach 17 at.%. Apparently, overheating of the melt and melting time during vacuum induction melting were insufficient to completely dissolve the Re. Even though the melting temperature cannot be measured, it is clear that it is well below the melting point of Re, i.e., 3181 °C, so that an extended time period is required to steadily dissolve Re. Further experiments revealed that within the limits of the available VIF furnace, only 15 at.% of Re can be reliably dissolved. For this reason, the target composition of the second melt, produced in-house, was adjusted to Co-15Re-5Cr, while the third melt was produced externally, as mentioned above, using a more powerful vacuum induction furnace.

Every composition of all alloys presented in this work was measured by EDX with a Hitachi TM3000 SEM equipped with the Bruker EDS-System Quantax 70 (Berlin, Germany). Due to the difficulties of measuring the exact carbon content with EDX, only the non-carbon elements were quantitively determined. Since the non-carbon elements only make up a fraction α of the overall amount of elements—assumingly, α = 100 at.% minus the nominal C-content in at.%—the measured contents of the non-carbon elements were weighted by the factor of α. All chemical compositions after EDX measurement depict the Re, Cr, Ta, Ti, and Hf weighted by factor of α. Hereby, it is assumed that the C-content is equal to the targeted value of the nominal composition. Throughout this study, this measured composition is given for all alloys. All heat treatments, were conducted in the Linn High Therm KKH-200/200/350/1600 Moly vacuum furnace (Eschenfelden, Germany). For the solution treatment, all investigated alloys were held at 1350 °C/5 h + 1400 °C/5 h + 1450 °C/5 h and quenched with Ar gas subsequently. Additional solution treatment at 1400 °C/60 min + 1500 °C/20 min and Ar gas quenching were conducted for some alloys in the same vacuum furnace. The cooling rate was measured by a thermocouple inside the vacuum furnace. For higher temperatures at around 1400 °C to 1300 °C the cooling rate is rather high at approx. 125 °C/min but slows down to approx. 60 °C/min at 1000 °C. The aging treatment was carried out in the same vacuum furnace without subsequent quenching. The X-ray diffraction (XRD) measurements were performed with a GE Inspection Technologies Diffractometer System XRD 3003 TT (Ahrensberg, Germany) using Cu K_α_ radiation (λ = 1.54056 Å).

## 3. Results

### 3.1. As-Cast Microstructure of Co-Re-Cr, Co-Re-Cr-Ta-C, Co-Re-Cr-Ti-C, and Co-Re-Cr-Hf-C

The microstructures of Co-10.7Re-5.3Cr in Figure 1 as well as Co-13.9Re-5.0Cr-1.5Ta-1.2C, Co-10.8Re-5.3Cr-1.3Ti-1.2C, and Co-11.6Re-5.3Cr-0.5Hf-1.2C in Figure 2 exhibit a similar distribution of Re in their as-cast condition. The contrast obtained by imaging with backscattered electrons (BSE) clearly reveals segregation of the heavy element Re towards dendrite cores and arms.

Similarly, the precipitates in the interdendritic regions of Co-Re-Cr-Ta-C and Co-Re-Cr-Hf-C also appear bright and indicate phases with heavy elements (Figure 2b,f). Adjacent to the bright heavy element phases, Co-Re-Cr-Hf-C displays darker areas Figure 2f). The precipitates in Co-Re-Cr-Ti-C are even darker and indicate a phase composition of light elements (Figure 2d). The precipitates of Co-Re-Cr-Ta-C and Co-Re-Cr-Hf-C form connected regions, often with an individual size larger than 10 µm, whereas the ones in Co-Re-Cr-Ti-C are disconnected, evenly distributed, and range below 10 µm.

### 3.2. Solution Treated and Aged Microstructure of Co-Re-Cr-Ta-C, Co-Re-Cr-Ti-C and Co-Re-Cr-Hf-C

The microstructures of Co-14.8Re-4.9Cr-1.9Ta-1.8C in Figure 3a and Co-14.6Re-5.0Cr-2.1Ti-1.8C in Figure 3c after ST indicate that a precipitation of fine particles has already occurred after quenching. Some large blocky TaC-particles within the matrix or at grain boundaries are also observed. Apparently, they were not dissolved during ST. However, the majority of the precipitates are either nano-sized needle-like (elongated) particles or particles with a globular shape. A similar morphology and size range are detected for the particles of Co-14.6Re-5.0Cr-2.1Ti-1.8C. Aging at 900 °C/15 h does not result in a noticeable coarsening of the particles, as seen in Figure 3b,d. Moreover, it is worth mentioning that for both alloys, an inhomogeneous distribution of brightness of the matrix even within a singular grain is observed, which might either indicate a subgrain structure or locally varying Re-content.

The BSE image in Figure 4a illustrates that within the Co-Re-Cr-Hf-C system, different phases are present after solution treatment, measuring in size already above 2 µm. The distinct mass contrast of the phases marked by the letters A to D in comparison to the surrounding matrix indicates the enrichment of large amounts of heavy elements in these phases. In this case, the depletion of Re, seen in Figure 4c, and the pronounced appearance of Hf in Figure 4e suggest some kind of Hf-based phase. The phase marked by the letter A is especially rich in Hf and C with a complete absence of Co, as depicted in Figure 4e,f, and b, respectively. This element distribution provides strong evidence for a Hf-C phase. The other phases marked by the letters B, C, and D show an overall high content of Hf and Co and a remarkable difference in brightness due to the mass contrast in the BSE image in Figure 4a. Nonetheless, these phases only differ slightly in their amounts of Co and Hf and do not display any difference at all for Re, Cr, and C. Possibly, the signals of Re and Cr are merely contributions of the underlying matrix or preparation artifacts, as in the case of C. This leads to the conclusion that phases B, C, and D may be different Co-Hf phases.

A local EDX measurement at the locations of phases B, C, and D revealed a nominal composition as indicated in Table 1. Taking the Co-Hf phase diagram into consideration [23], it becomes evident that the phases B, C, and D must be attributed to one or several of the following phases: Co_7_Hf, Co_23_Hf_6,_ and Co_7_Hf_2_. Although the content of Co and Hf for the phases C and D is in good accordance with the nominal compositions of Co_23_Hf_6_ and Co_7_Hf_2_, it is not possible to clearly distinguish the phases due to the marginal differences in composition. Small contributions from the surrounding phases may affect the absolute values regarding the Co- and Hf-contents. The same reasoning applies to the phase marked by the letter B, where an unequivocal attribution neither to one of the lower Co-content phases (Co_23_Hf_6_ and Co_7_Hf_2_) nor to the higher Co-content phase (Co_7_Hf) might be justified. 

An X-ray diffraction analysis of Co-13.5Re-3.2Cr-2.3Hf-2.4C in its solution-treated state, as seen in Figure 5, does not provide any clarity on this topic. The low volume fraction of the Co-Hf phases in comparison to the Co-Re-Cr matrix only allows the detection of their most pronounced peaks, which all coincide between 44° and 45°. Any other isolated peak for the mentioned Co-Hf phases cannot be detected, as is the case for the weak peak of HfC at 33.4°. Moreover, it is observed that the strongest peak measured at 46.5° deviates from the literature value of hcp-Co. This shift to lower diffraction angles has already been observed in [14] and is caused by a lattice expansion induced by dissolved Re within hcp-Co.

Contrary to Co-Re-Cr-Ta-C and Co-Re-Cr-Ti-C, only a small amount of HfC precipitates exist as nano-sized particles within the lath-like matrix after solution treatment and aging at 1000 °C/15 h of Co-12.9Re-4.1Cr-0.5Hf-1.2C, as seen in Figure 6. The lath-like microstructure indicates a hcp ⇄ fcc-transformation during aging and subsequent cooling. An EDX measurement in this area revealed a composition with approx. 11 at.% Re, 4.2 at.% Cr, and 0.06 at.% Hf. In direct proximity to the lath-like microstructure, a large area of Co-matrix without a lath-like microstructure is located where the Re-, Cr-, and Hf-contents are 12.2 at.%, 5.1 at.%, and 0 at.%, respectively. The extremely low (in the lath-like matrix) and non-existent (in the matrix without laths) Hf-content provides further confirmation of the absence of any or an insignificant amount of precipitates after aging. Apparently, most of the Hf is consumed by the larger Hf-containing particles.

### 3.3. Influence of Re and Cr on the Microstructure and Precipitation in Co-Re-Cr, Co-Re-Cr-Ta-C and Co-Re-Cr-Ti-C

As already mentioned in the introduction, during the investigation of carbide-strengthened Co-17Re-5Cr-based alloys, the authors encountered difficulties reproducing a Co-Re-Cr matrix that completely transforms to fcc-Co during the solution treatment. This becomes evident from the contemplation in Figure 7 and Figure 8a,b, where the solution-treated microstructures of the carbide-free (Co-16.5Re-4.8Cr) and TaC-strengthened (Co-16.8Re-4.5Cr-2.4Ta-1.8C) Co-17Re-5Cr alloys are depicted, respectively. These microstructures feature large areas of Re-rich grains, which appear as bright grains due to the mass contrast in the BSE images, as seen in Figure 7a,c,d. The element mapping of Re in Figure 7b and the local EDX measurements within and outside the Re-rich grains undoubtedly express the differences in the Re-distribution across the microstructure. Re-rich areas show a Re-content of approx. 20 at.%, whereas Re-poor areas are approx. 14.8 at.%, while the Cr-content ranges close to the nominal composition of 5 at.%.

Figure 8a,b display that within the matrix of alloy Co-16.5Re-4.5Cr-2.4Ta-1.8C, the Re-rich grains do not contain any TaC. Instead, TaC precipitates as coarse particles along the grain boundaries as well as fine and blocky particles within the supposedly fcc-Co matrix (during ST) after quenching, as seen in Figure 8a,b. This can only happen if the Re-rich grains exhibit a hcp crystal structure during ST, as the solubility of Ta in hcp-CoRe is very low, so that Ta partitions towards the fcc grains during ST, leaving Ta-depleted hcp grains behind. The reduction of the hcp-stabilizing Cr-content from 5 at.% to 0 at.% does not eradicate the TaC-free grains in Co-16.1Re-2.1Ta-1.8C since multiple Re-rich grains are still visible in Figure 8c. Similar to Co-16.1Re-2.1Ta-1.8C, Co-16.4Re-2.0Ti-1.8C yields the same outcome as TiC precipitates cannot be observed within the Re-rich grains. Instead, coarse particles result along the grain boundaries due to the poor solubility of Ti in hcp-CoRe [24].

Another attempt to eradicate the presence of hcp during solution treatment was conducted by additional heat treatment at 1400 °C/60 min + 1500 °C/20 min for Co-16.0Re-4.8Cr (Figure 9a) and Co-16.5Re-4.8Cr-1.9Ta-1.8C (Figure 9b) after initial ST. Both alloys display a significant amount of Re-rich grains, whereas Co-16.8Re-4.5Cr-2.4Ta-1.8C shows a comparatively smaller area fraction of those Re-rich grains. 

The authors did not consider a further increase above 1500 °C because the solidus temperature of Co-17Re [6] (and thereby of Co-16.5Re-4.8Cr and Co-16.1Re-2.1Ta-1.8C, probably) ranges slightly above 1500 °C so that higher temperatures bear the risk of incipient melting. Instead, a step-wise reduction of the hcp-stabilizing element Re was realized by the targeted alloys of Co-16Re, Co-15Re-5Cr-1.8Ta-1.8C, and Co-15Re-5Cr-1.8Ti-1.8C. For Co-15Re-5Cr-1.8Ta-1.8C and Co-15Re-5Cr-1.8Ti-1.8C, the amount of Cr was kept at the initial value of 5 at.% for the following reasons: Cr is supposed to prevent the decrease of the fcc-Co start temperature, thus hindering the occurrence of retained fcc-Co after cooling from ST. In-situ neutron diffraction measurements of Co-17Re-5Cr-1.2Ta-1.082C and Co-17Re-1.2Ta-1.082C revealed a remarkable decrease of the fcc-Co start temperature by approx. 160 °C and the presence of retained fcc-Co after ST, induced by the removal of Cr [17]. A low fcc-Co start temperature reduces the ultimate application temperature of the alloy because (i) TaC is not stable in the fcc phase and (ii) the fcc phase is weaker than the hcp phase. Figure 10 reveals that approx. 15.7 at.% of Re is still sufficient to produce a Co-Re matrix that contains Re-rich grains and thereby hcp-Co during ST. 

The authors assumed that Co-16Re-TaC and Co-16Re-TiC alloys would yield the same result as was observed for the targeted compositions of Co-17Re-5Cr and Co-17Re-5Cr-1.8Ta-1.8C in Figure 7 and Figure 8, respectively. Therefore, TaC- and TiC-containing alloys were cast with a Re-content of 15 at.%. Figure 11 illustrates the microstructures of Co-15.3Re-4.9Cr-1.8Ta-1.8C and Co-15.1Re-5.1Cr-2.3Ti-1.8C after ST and aging at 1100 °C/15 h. Both microstructures do not display any Re-rich grains where no carbides precipitate but rather a homogeneous Re-distribution expressed by a uniform color distribution in the matrix. Consequently, it is assumed that no hcp-Co is present during ST. Furthermore, no lath-like structures are detected after aging at 1100 °C, which would indicate the presence of the fcc phase during aging. The partial phase transformation from hcp-Co to fcc-Co on heating yields a characteristic lath-like structure that is easily identified at room temperature after the fcc phase is transformed back to the hcp phase. Former fcc regions contain less Re and appear less bright in direct comparison to the adjacent regions, which remained hcp during aging. The XRD measurement of Co-14.7Re-5.1Cr in Figure 5 shows the existence of isolated hcp-Co peaks at approx. 41° and 47°, whereas an overlap for fcc-Co and hcp-Co occurs at approx. 44°. However, no fcc-Co peak at 51° is observed.

### 3.4. Solubility of TaC and TiC in Co-Re-Cr

Figure 12 depicts the microstructure of the Co-Re-Cr matrix with varying xTa-xC- and xTi-xC-contents with x = 1.2, 1.8, and 2.4. Figure 12 a provides an overview of Co-16.6Re-4.7Cr-1.5Ta-1.2C, where neither fine nor coarse TaC precipitates are visible after ST. Despite the fact that the matrix composition of Co-17Re-5Cr tends to retain a certain amount of hcp-Co during ST, in which Ta and C do not dissolve (as seen in Figure 8), the reduced amount of available fcc-Co does not seem to lead to an oversaturation of fcc-Co. On the contrary, the first coarse TaC particles appear in the composition of Co-14.8Re-4.9Cr-1.9Ta-1.8C in Figure 12b, and even larger populations of coarse TaC emerge for Co-16.3Re-4.8Cr-2.7Ta-2.4C Figure 12c. The observations from Figure 11 suggest that no hcp-Co grains remain during ST for a matrix composition containing less than or equal to 15 at.% Re, so that a complete fcc-Co matrix is available for solving Ta and C, presumably. Thus, the presence of a small amount of coarse TaC particles in the alloy Co-14.8Re-4.9Cr-1.9Ta-1.8C suggests that the solubility limit of the fcc phase for Ta, C, at 1450 °C is slightly exceeded. Similar observations are made for Co-Re-Cr-TiC, where coarse TiC precipitates are absent for Co-10.8Re-5.2Cr-1.2Ti-1.2C, as seen in Figure 12d. However, Figure 12e shows the appearance of coarse and even Chinese-script-like (CS-like) TiC particles for Co-14.6Re-5.0Cr-2.1Ti-1.8C. The formation of CS-like TiC becomes even stronger for Co-13.4Re-4.6Cr-2.9Ti-2.4C, where large networks of CS-like TiC are observed (see arrows in Figure 12f). Note that CS-like morphologies were not found in the case of TaC.

## 4. Discussion

### 4.1. As-Cast Microstructure of Co-Re-Cr, Co-Re-Cr-Ta-C, Co-Re-Cr-Ti-C and Co-Re-Cr-Hf-C

The Re-rich dendritic structure of Co-17Re-5Cr is typical for Co-Re-based alloys and has already been observed in the cast microstructures [3]. The element mapping results from the EDS measurements in [17] show that Cr and Co segregate into the interdendritic regions, where also Ta- or Cr-carbides form. The as-cast microstructures of Co-Re-Cr-Ta-C, Co-Re-Cr-Ti-C, and Co-Re-Cr-Hf-C in Figure 2 exhibit the same solidification behavior as the corresponding precipitates, which similarly appear in the interdendritic regions. Particularly, Co-Re-Cr-Ta-C and Co-Re-Cr-Hf-C feature bright precipitates, which suggest the existence of heavy element phases, likely Ta- and Hf-rich phases such as TaC and HfC. In comparison, Figure 2b depicts darker precipitates in Co-Re-Cr-Ti-C, indicating light element phases, presumably TiC. In the case of Co-Re-Cr-Hf-C, it is unclear whether the darker areas represent another carbide phase containing light elements or matrix areas depleted of heavy elements.

### 4.2. Solution Treated and Aged Microstructure of Co-Re-Cr-Ta-C, Co-Re-Cr-Ti-C and Co-Re-Cr-Hf-C

The precipitation of nano-sized TaC after ST, displayed in Figure 3, matches the findings in [16] for the compositions of Co-17Re-1.2Ta-1.082C and Co-17Re-15Cr-1.2Ta-1.082C. Upon cooling, the transformation from fcc-Co to hcp-Co completes at the hcp-Co finish temperature. This abruptly decreases the solubility of Ta and C within the matrix and starts the precipitation from the oversaturated matrix. The existence of a hcp-Co finish temperature is an absolute necessity for carbide-strengthened Co-Re(-Cr) alloys because otherwise fcc-Co would be retained. Then, TaC would dissolve in fcc-Co, and, furthermore, fcc-Co would provide less creep strength as it is significantly softer than hcp-Co. Both mechanisms are softening mechanisms during creep and, thus, are undesirable. So far, no literature on the precipitation behavior of TiC within the Co-Re-Cr matrix has been published. The fact that TiC precipitation from the oversaturated matrix after ST in Figure 3 appears quite similar in size and precipitated volume fraction to the one of TaC suggests that nucleation and growth behavior may be similar, too. The shape of the TaC precipitates is either finely globular or needle-like in Figure 11a, which is in good accordance with the findings in [15]. The needle-like shape may be a consequence of the fcc-Co to hcp-Co phase transformation, which propagates as laths throughout the matrix. First, TaC nuclei might form at the lath boundaries and grow along them to some extent, eventually resulting in an elongated shape. As for TiC, the same observation is made in Figure 11c, giving another hint of similar precipitation behavior. In contrast, Co-13.5Re-3.2Cr-2.3Hf-2.4C neither exhibits a precipitation of a single precipitate phase nor are the precipitate sizes in the nanometer range but rather several µm. The XRD diagram from Figure 5 clearly confirms the existence of a HfC phase because of the isolated diffraction peak at 33.4°. The weakness of the diffraction peak emphasizes the low volume fraction of HfC. The element mapping from Figure 4 and the local compositions at the positions C and D (see Table 1) suggest either Co_23_Hf_6_ or Co_7_Hf_2_. However, it is not possible to unequivocally determine the phases since both phases exhibit a very similar theoretical composition. Small inaccuracies in EDX measurements due to signals from the surrounding matrix may result in a misattribution of the phases. The same reasoning can be applied for the phase at position B, where 17.5 at.% Hf is measured. This Hf-content lies in between the theoretical composition of the high Hf phases Co_23_Hf_6_ or Co_7_Hf_2_ and the low Hf phase Co_7_Hf. The XRD diagram does not provide any evidence of the true nature of the phases at positions C and D either since the strongest diffraction peaks of both phases coincide at approx. 44° and are superimposed by a hcp-Co peak. Unfortunately, the second-strongest peaks at 40.8° and 37.2° for Co_23_Hf_6_ and Co_7_Hf_2_, respectively, cannot be detected because of the comparatively low volume fraction of these phases. The aged microstructure of Co-Re-Cr-Hf-C at 1000 °C/15 h, presented in Figure 6, resulted in an insignificant amount of additional precipitation of nano-sized HfC or the suggested Co-Hf phases, which appear only in the areas of the lath-like microstructure. Instead, the lath-free areas of the microstructure are characterized by the absence of such nano-sized particles and the complete lack of Hf. This is consistent with the elemental distribution of Hf in Figure 4e after ST. It illustrates a very low content of Hf in the matrix, as the majority of Hf is bound either as coarse HfC or as large Co-Hf phases. Thereby, no or a negligible amount of Hf is present for the precipitation of fine particles during aging. In general, coarse particles will contribute less to the creep resistance due to the large interparticle spacing regardless of the bypass mechanism, be it climb, detachment, or the Orowan mechanism. The existence of a lath-like microstructure and matrix areas without it indicate that aging at 1000 °C takes place very close to the fcc-Co start temperature, where small changes in the content of the hcp-stabilizing elements Re and Cr determine whether the matrix stays hcp-Co or (partially) transforms during aging.

### 4.3. Influence of Re and Cr on the Microstructure and Precipitation in Co-Re-Cr, Co-Re-Cr-Ta-C and Co-Re-Cr-Ti-C

The observations from Figure 7 and Figure 8 provide clear evidence that a Co-Re-Cr matrix containing approx. 17 at.% Re and 5 at.% Cr does not result in a complete fcc-Co matrix at 1450 °C. Instead, the in-situ neutron diffraction measurements from [17] show that for Co-17Re-5Cr-1.2Ta-1.082C (with y = 0.9), which is very close to the composition of Co-17Re-5Cr-1.2Ta-1.2C (with y = 1), a complete phase transformation to fcc-Co takes place at 1410 °C, upon heating. Although the study of [17] suggests that the fcc-Co finish temperature depends on the C/Ta-ratio, where a drop of approx. 100 °C comparing y = 0.5 to y = 0.9 was measured, it is unlikely that the minor difference regarding the C/Ta-ratio of y = 0.9 and y = 1 is responsible for this discrepancy. Therefore, it is unreasonable to assume that the stochiometric C/Ta-ratio in this study may have significantly shifted the fcc-Co finish temperature to higher temperatures even above 1500 °C, as indicated by the Re-rich grains for the carbide-free and TaC-strengthened Co-17Re-5Cr matrix in Figure 9a and 9b, respectively. However, the findings of this study and the aforementioned study [17] concerning the influence of Cr on the fcc-Co finish temperature are in good accord. The reduction of Cr from 5 at.% to 0 at.% did not significantly lower the fcc-Co finish temperature, as proven by the Re-rich/TaC- and TiC-free grains in Figure 8c,d. Thus, Re is regarded as the decisive factor determining the fcc-Co finish temperature, and a complete hcp-Co to fcc-Co transformation under ST conditions is successfully reached at approx. 15 at.% Re (see Figure 11) rather than 16 at.% Re (see Figure 10). This observation better corresponds with the Co-Re phase diagram, where Co-15Re is still located within the two-phase region of hcp-Co and fcc-Co at 1450 °C but is very close to the single-phase fcc region, which is reached at slightly higher temperatures in a narrow temperature range. In comparison, Co-16Re and Co-17Re exhibit, up to the solidus temperature above 1500 °C, only a two-phase region of fcc-Co and hcp-Co [6]. It is unclear from exactly where the discrepancy concerning the fcc-Co finish temperature for Co-17Re-5Cr-1.2Ta-1.082C in [17] and for Co-16.6Re-4.7Cr-1.5Ta-1.2C (as seen in Figure 12a) in this study stems. It is suspected that the melting procedure might be the source of error since the very same vacuum induction furnace was used in both studies and the first master alloy of this study with the nominal composition of Co-17Re-5Cr failed to reach 17 at.% (as described in the introduction). Furthermore, previous studies on VAM-melted Co-17Re-based alloys that checked the chemical composition revealed a common fluctuation of Re in the range of a few at.% compared to the targeted composition. Specifically, the Re-content in the Co-Re-matrix was identified in [25] as 14.6 at.% and 16.6 at.%, in [12] as 14.8 at.% and 15.6 at.%, and in [26] as 17.7 at.%, while the targeted amount was 17%. Apparently, the exact incorporation of the targeted amount of Re and its homogeneous distribution across the matrix are difficult to achieve either with VIF or VAM (and the subsequent solution heat treatment). To ensure a single-phase region of fcc-Co at 1450 °C, it is therefore recommended to select 15 at.% of Re. Another point worth discussing is the question of possibly retaining fcc-Co after ST. Although in Figure 5 for Co-15Re-5Cr, the strongest fcc-Co peak coincides at 44° with another hcp-Co peak, the second-strongest peak at 51° does not emerge. This indicates a rather low amount or the complete absence of retained fcc-Co. This agrees with the findings in [17], where retained fcc-phase was not observed after cooling from ST in the case of the Cr-containing Co-Re-alloys. Taking into consideration that mainly off-stochiometric TaC compositions in Co-Re-Ta-C-alloys (without Cr) with C/Ta-ratios lower than 0.9 result in a larger volume fraction of retained fcc-Co due to the fcc-stabilizing nature of free Ta [15], the likelihood of retained fcc-Co in Co-15Re-5Cr-1.8Ta-1.8C and Co-15Re-5Cr-1.8Ti-1.8C, exhibiting a C/Ta-ratio of one, is rather low. Particularly, the weak fcc-stabilizing effect of Ti compared to Ta [27] makes any significant amounts of retained fcc-Co in the Ti-containing alloys unlikely.

### 4.4. Solubility of TaC and TiC in Co-Re-Cr

The microstructures of Co-16.6Re-4.7Cr-1.5Ta-1.2C (Figure 12a) and Co-10.8Re-5.2Cr-1.2Ti-1.2C (Figure 12c) confirm that after quenching, no coarse TaC or TiC precipitates exist, and therefore no oversaturation of the fcc-Co matrix with Ta/Ti and C occurs during ST. However, this is the exact case for Co-14.8Re-4.9Cr-1.9Ta-1.8C (Figure 12b) and Co-14.6Re-5.0Cr-2.1Ti-1.8C (Figure 12d) and becomes even more pronounced for higher Ta-, Ti-, and C-contents (Figure 12c,e). Therefore, it is reasonable to assume that the solubility limits of xTa-xC and xTi-xC for Co-15Re-5Cr lie somewhere between x = 1.2 at.% and 1.8 at.%. Note that the observed solubility limits pertain to C/Ta- and C/Ti-ratios of one (y = 1). They might differ for other ratios. In fact, the isothermal section of the ternary Co-Ti-C system at 1250 °C in [28] shows that the Ti-solubility steadily increases as the C/Ti-ratio decreases, while the opposite is true for the C-solubility. Assuming that the same holds true for the Co-Re-Cr-Ti/Ta-C system under investigation here, y = 1 is the best choice to allow for the highest possible amount of MC-precipitation (M: Ta, Ti) in the hcp-matrix. For y ≠ 1 either the maximum soluble amount of M or C would diminish below the value at y = 1, thus reducing the possible amount of MC-precipitation.

The contribution of Cr and Re to the solubility limits of Ta and Ti within fcc-Co is difficult to estimate. An experimental study of the Co-Re-Ta system indicates that fcc-Co-producing compositions such as Co-11Re-9Ta and Co-16Re-10Ta show a reduced solubility of Ta at 1300 °C within fcc-Co as the Re-content rises. However, Re raises the solidus temperature in combination with Co (different from the binary systems of Co-Ta, -Cr [6], -C [29], and -Ti [24]), making solution heat treatments at higher temperatures for higher solubility of Ta possible. Both effects of Re on the solubility of Ta might cancel each other out eventually. To the author’s best knowledge, no literature on the Co-Re-Ti system is available, so the effect of Re on the solubility of Ti in fcc-Co is unknown. The addition of up to approx. 10 at.% of Cr to the Co-Ta- or Co-Ti system does not significantly reduce the solubility of Ta [30] or Ti [31] in fcc-Co. Thus, there is no evidence in the contemporary literature that the solubility limits of Ta and Ti within the fcc-Co-Re-Cr matrix might be significantly affected by small changes in the Re- and Cr-content.

Altogether, the compositions of Co-15Re-5Cr-1.8Ta-1.8C and Co-15Re-5Cr-1.8Ti-1.8C pose a good choice to further investigate and assess carbide-strengthened Co-Re(-Cr) alloys for high-temperature applications. Smaller amounts of coarse TaC and TiC might locally lead to the depletion of fine precipitates after aging and will not, by themselves, improve the creep resistance in a significant way. However, it is assumed that these aspects are neglectable in comparison to the effect of a completely saturated fcc-Co matrix during ST and eventually a higher volume fraction of fine precipitates in the hcp-Co matrix.

## 5. Conclusions

This study provides clear evidence that Co-Re(-Cr)-based alloys with a Re-content larger than 15 at.% do not exhibit a complete fcc-Co microstructure during solution treatment with a maximum annealing temperature of 1450 °C. Instead, a mixture of fcc-Co and hcp-Co is present, which is visible at ambient temperature as a distribution of Re-poor (fcc-Co during ST) and Re-rich grains (hcp-Co during ST). Contrary to the Re-poor grains, the Re-rich grains evidently show a complete lack of TaC and TiC after ST due to the low solubility of Ta and Ti in hcp-Co and the partitioning of these elements towards fcc-Co during ST. An increase of the maximum temperature for the solution treatment up to 1500 °C does not result in complete transformation of hcp-Co to fcc-Co either, as Re-rich grains are still visible for the carbide-free alloy (Co-17Re-5Cr) and the TaC-strengthened alloy (Co-17Re-5Cr-1.8Ta-1.8C). Despite the fact that retained fcc-Co at ambient temperature cannot be unambiguously ruled out by XRD measurements, the likelihood of its existence in Cr-containing Co-Re-alloys such as Co-15Re-5Cr or carbide-strengthened Co-15Re-5Cr alloys with a stochiometric composition of the carbide-forming elements (e.g., Ta-C and Ti-C) is considered to be very low. Moreover, aging at 1100 °C retains the hcp-Co matrix because no lath-like structures due to hcp ⇄ fcc-transformation are found. Therefore, all four requirements from the introduction are fulfilled by Co-15Re-5Cr. Furthermore, TaC and TiC are excellent carbide options as these precipitate as nano-sized particles, which is predominantly not the case for HfC. The maximum solubility of xTa-xC and xTi-xC is found to be above x = 1.2 but somewhat less than x = 1.8. A minor oversaturation of the fcc-Co matrix leading to the precipitation of a few coarse TaC and TiC particles after ST is considered advantageous as it allows the solubility of the matrix for Ta and C to be fully utilized and, ultimately, a maximum content of fine precipitates to be achieved. For these reasons, Co-15Re-5Cr-1.8Ta-1.8C and Co-15Re-5Cr-1.8Ti-1.8C are good choices for conducting a comprehensive study on the creep properties of carbide-strengthened Co-Re-based alloys.

## Figures and Tables

**Figure 1 materials-16-04443-f001:**
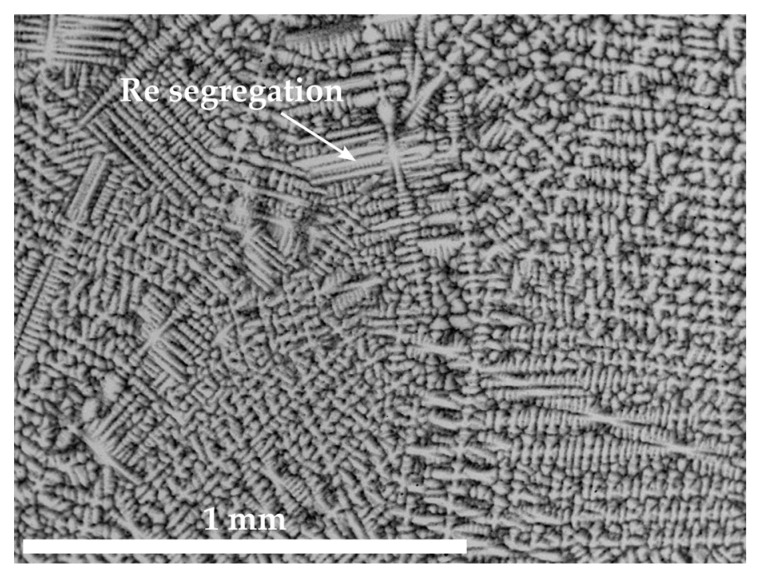
The BSE image of Co-10.7Re-5.3Cr in its as-cast condition shows Re enrichment in the dendrite cores and arms.

**Figure 2 materials-16-04443-f002:**
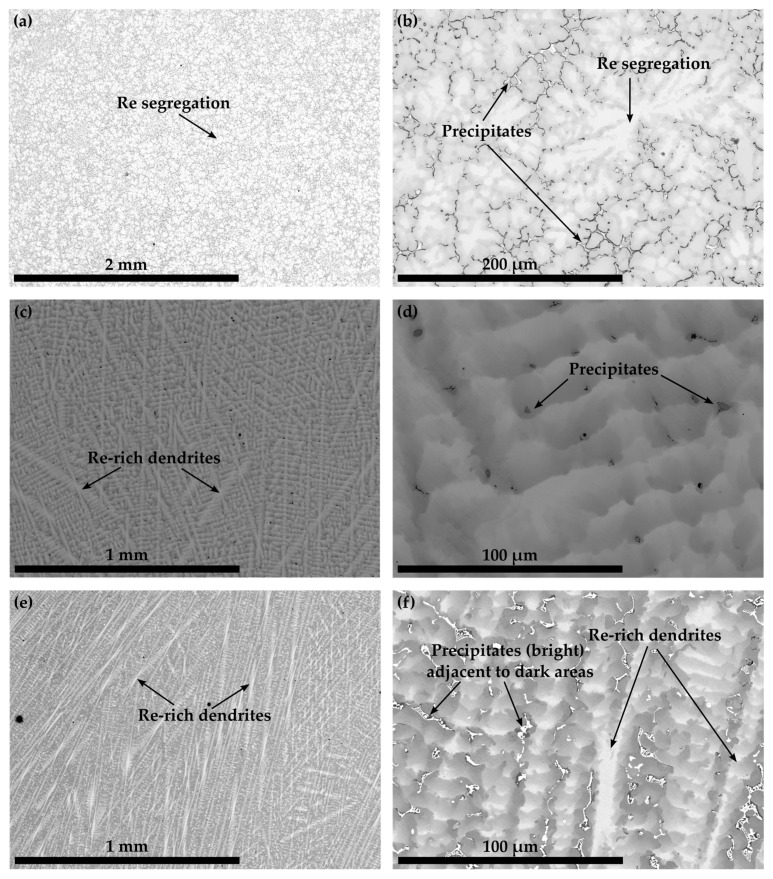
BSE images displaying the as-cast microstructures of Co-13.9Re-5.0Cr-1.5Ta-1.2C in (**a**,**b**), Co-10.8Re-5.3Cr-1.3Ti-1.2C in (**c**,**d**), and Co-11.6Re-5.3Cr-0.5Hf-1.2C in (**e**,**f**). Pronounced dendritic segregation of Re and precipitates within the interdendritic regions can be identified.

**Figure 3 materials-16-04443-f003:**
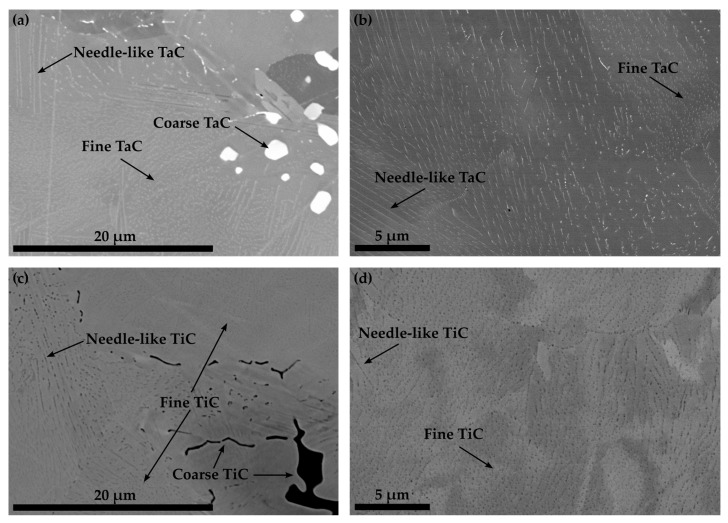
BSE images illustrating the solution treated microstructure of Co-14.8Re-4.9Cr-1.9Ta-1.8C in (**a**) and Co-14.6Re-5.0Cr-2.1Ti-1.8C in (**c**), whereas the microstructure subsequently aged at 900 °C/15 h is presented in (**b**,**d**), respectively. Nano- and micro-sized TaC and TiC precipitates are present after ST.

**Figure 4 materials-16-04443-f004:**
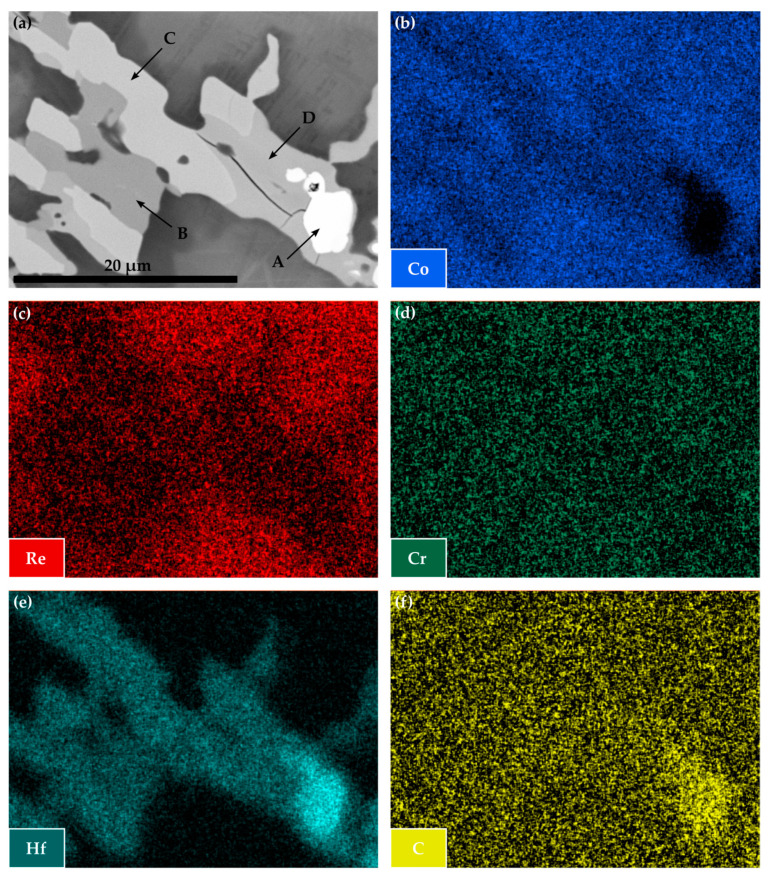
BSE image of Co-13.5Re-3.2Cr-2.3Hf-2.4C in its solution treated state in (**a**) and the corresponding element mapping images of Co, Re, Cr, Hf, and C in (**b**–**f**), respectively. The letters from A to D mark zones where the respective local composition was measured by EDX.

**Figure 5 materials-16-04443-f005:**
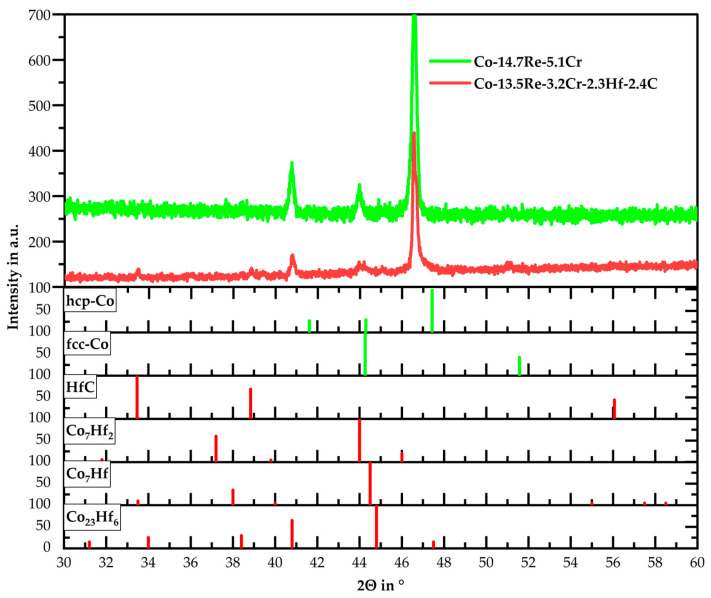
Comparison of the XRD patterns of hcp-Co, fcc-Co, HfC, Co_7_Hf, Co_7_Hf_2,_ and Co_23_Hf_6_ with the measured XRD patterns of solution treated Co-14.7Re-5.1Cr and Co-13.5Re-3.2Cr-2.3Hf-2.4C. All XRD patterns originate from the ICDD database.

**Figure 6 materials-16-04443-f006:**
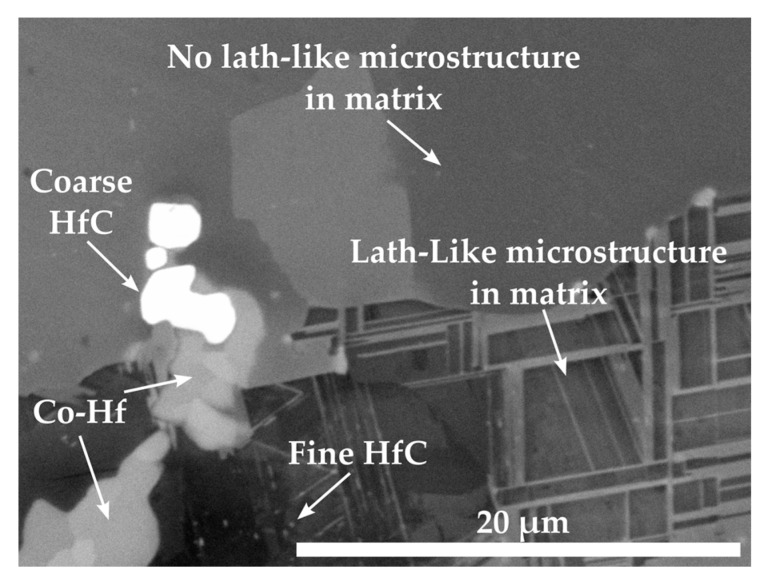
BSE image illustrating the solution treated and aged at 1000 °C/15 h microstructure of Co-12.9Re-4.1Cr-0.5Hf-1.2C. HfC and the Co-Hf phases appear as micro-sized precipitates and only a few nano-sized particles in the lath-like microstructure.

**Figure 7 materials-16-04443-f007:**
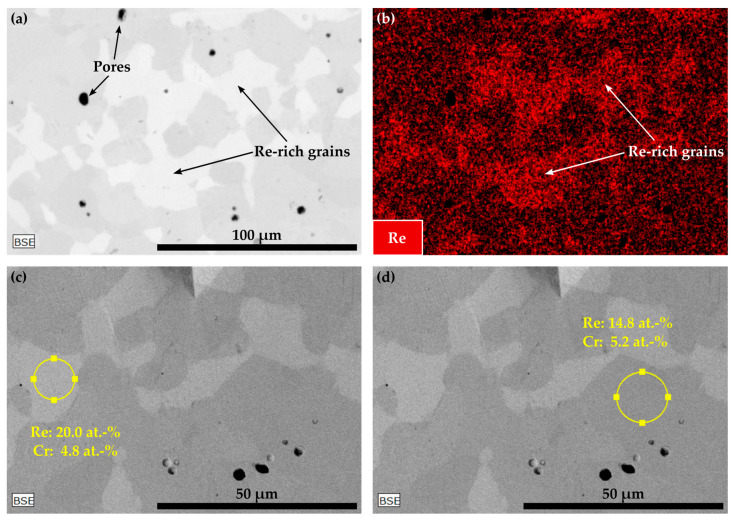
The microstructure of a solution treated Co-16.5Re-4.8Cr alloy: the BSE image in (**a**) depicts Re-rich grains, which is confirmed by the Re-distribution in (**b**) and local EDX measurements within and outside the brighter regions as seen in (**c**,**d**), respectively.

**Figure 8 materials-16-04443-f008:**
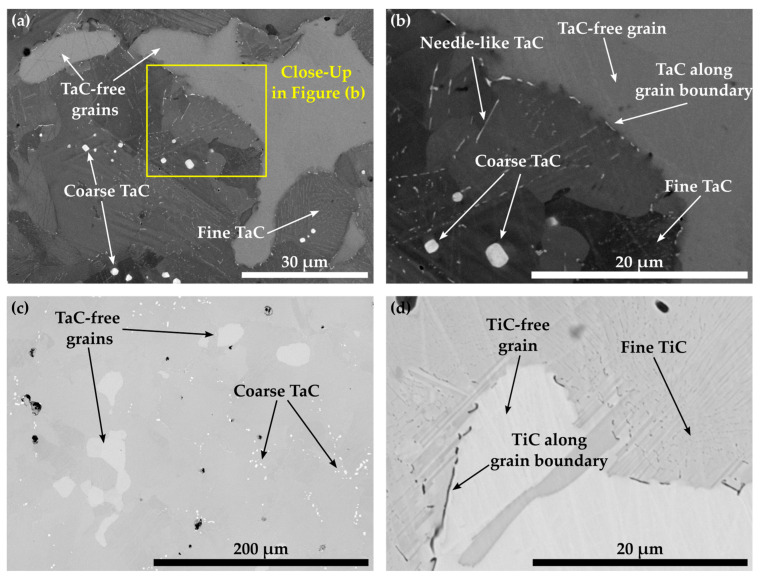
BSE images depicting the solution-treated microstructure of Co-16.8Re-4.5Cr-2.4Ta-1.8C in (**a**,**b**) as well as the solution-treated microstructure of Co-16.1Re-2.1Ta-1.8C and Co-16.4Re-2.0Ti-1.8C in (**c**,**d**), respectively. All stated alloys either containing 5 at.% (**a**,**b**) or 0 at.% of Cr (**c**,**d**) display Re-rich areas, free of any precipitates but forming coarse TaC and TiC along grain boundaries.

**Figure 9 materials-16-04443-f009:**
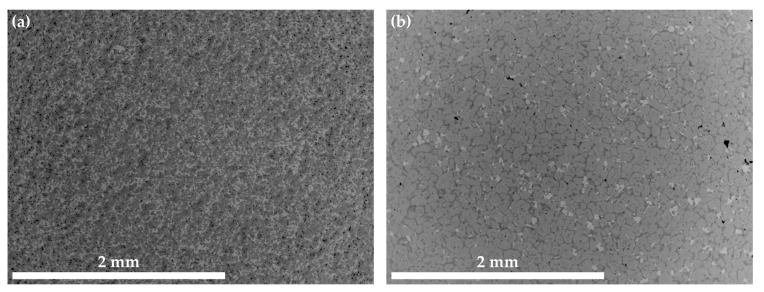
BSE images showing the microstructure of Co-16.5Re-4.8Cr in (**a**) and Co-16.8Re-4.5Cr-2.4Ta-1.8C (**b**) after ST and a subsequent heat treatment at 1400 °C/60 min + 1500 °C/20 min. The bright grey areas in both images represent the Re-rich grains.

**Figure 10 materials-16-04443-f010:**
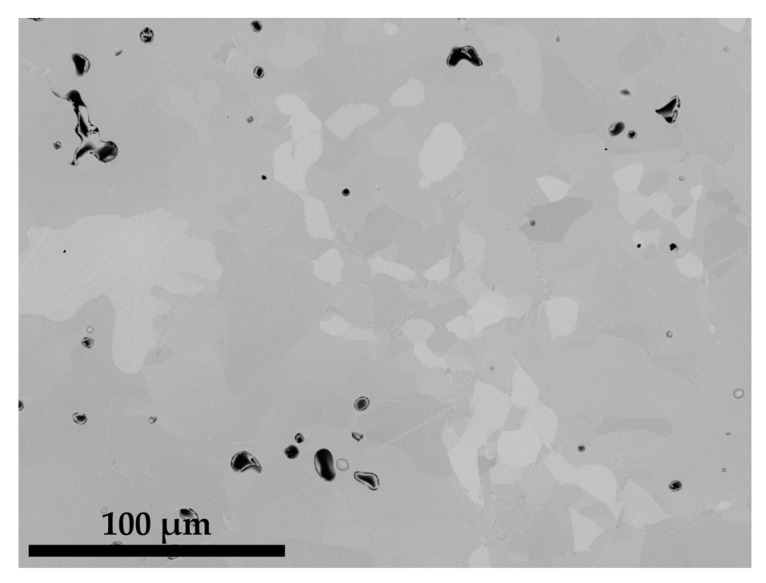
BSE image revealing a distribution of Re-rich grains for the Co-15.7Re alloy after ST. The bright grey areas represent the Re-rich grains.

**Figure 11 materials-16-04443-f011:**
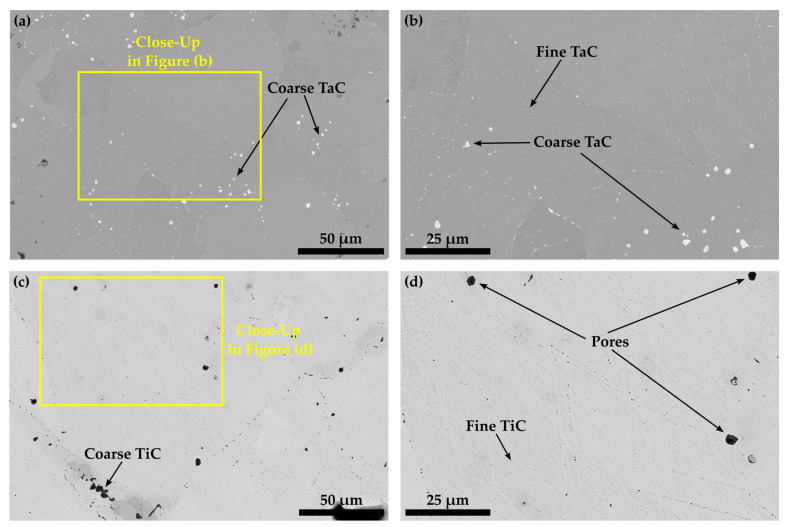
The microstructures of Co-15.3Re-4.9Cr-1.8Ta-1.8C in (**a**,**b**) and Co-15.1Re-5.1Cr-2.3Ti-1.8C in (**c**,**d**) after ST and aging at 1100 °C/15 h display neither Re-rich grains indicating the presence of hcp-Co during ST nor lath-like structures indicating the presence of fcc-Co during aging.

**Figure 12 materials-16-04443-f012:**
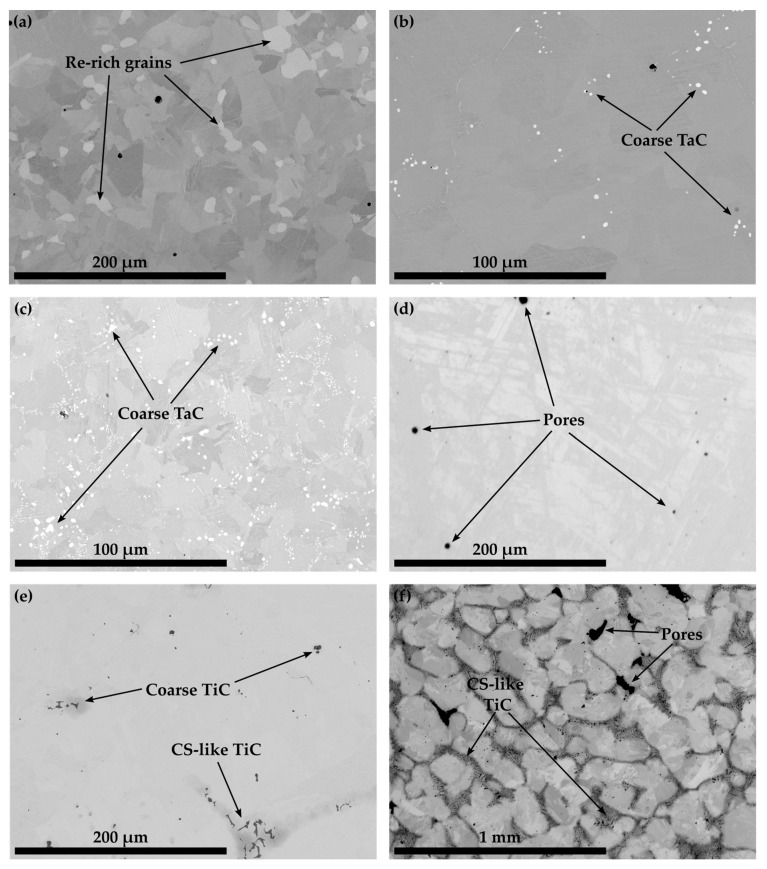
The BSE images illustrate the solution treated microstructure of Co-16.6Re-4.7Cr-1.5Ta-1.2C, Co-14.8Re-4.9Cr-1.9Ta-1.8C and Co-16.3Re-4.8Cr-2.7Ta-2.4C in (**a**–**c**) whereas (**d**–**f**) depict Co-10.8Re-5.2Cr-1.2Ti-1.2C, Co-14.6Re-5.0Cr-2.1Ti-1.8C and Co-13.4Re-4.6Cr-2.9Ti-2.4C, respectively.

**Table 1 materials-16-04443-t001:** Local Co- and Hf-contents measured at the locations of phases B, C, and D as well as the nominal compositions of intermetallic Co-Hf-phases from the Co-Hf phase diagram [23].

	Co [at.%]	Hf [at.%]		Co [at.%]	Hf [at.%]
B	82.5	17.5	Co_7_Hf	87.5	12.5
C	79.4	20.6	Co_23_Hf_6_	79.3	20.7
D	78.4	21.6	Co_7_Hf_2_	77.8	22.2

## Data Availability

Data available on request.

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
