# Peer review of "Reassessment of the Matrix Composition of Co-Re-Cr-Based Alloys for Particle Strengthening in High-Temperature Applications and Investigation of Suitable MC-Carbides"

_materials, 2023, doi:10.3390/ma16124443_

Round 1

Reviewer 1 Report

GENERAL COMMENTS

This is an interesting manuscript, which starts with a long but complete introduction efficiently presenting the studied problem. The discussion and conclusion parts are also strong. On the other hand, the Results part, which is very rich in illustrated microstructure descriptions, is a little too dense and it can be useful to divide this part into more than two paragraphs; one can also think to split the articles in two ones, for instance one treating the case of some monocarbides and one treating the case of the other monocarbides. The rather great length of the present manuscript (22 pages) allows such division.

QUESTIONS:

Top of page 4: in the two-step melting process, were there no some problems of flotation of the added graphite during the second step?

The Cr content is rather small (5wt.%); do you think that fast hot oxidation cannot be encountered? Are applications some protective coatings planned?

MINOR :

Line 73 : under-stochiometric®under-stoichiometric

Line 92: conditions: ®conditions.

Line 166 : place Instead ®place. Instead; etc … please check the entire manuscript and correct the found small problems.

Pages 5-7: the sizes of the pictures would be decreased for the final version of the manuscript; and the arrows seem to be a little shifted and they do not show what they ought to. In fact, this is the same for the most of figures. Figure 4 must be re-arranged?

Figure 5: the too high proximity of the numbers of the Y-axis (…255075100…) is unpleasant.

OTHERS:

The “Error! Reference source not found” is found several times in the text, revealing a problem of automatic linking. This is a little perturbing during reading for good understanding.

Some whole sentences are present in some figures captions: this should be avoided.

Only rare little problems found. However a re-reading by the authors can be useful.

Author Response

Dear reviewer,

we kindly thank you for the review of our article and highly appreciate the time you dedicated for giving us your feedback. We are very glad that you consider our work as interesting and gave us your recommendation for the publication of this article. We discussed your feedback and made according adjustments (highlieghted in blue letters) which you can find in the PDF attached to this message. This time we uploaded the draft as a PDF file in order to avoid any issues with the formation.

Point 1: This is an interesting manuscript, which starts with a long but complete introduction efficiently presenting the studied problem. The discussion and conclusion parts are also strong. On the other hand, the Results part, which is very rich in illustrated microstructure descriptions, is a little too dense and it can be useful to divide this part into more than two paragraphs; one can also think to split the articles in two ones, for instance one treating the case of some monocarbides and one treating the case of the other monocarbides. The rather great length of the present manuscript (22 pages) allows such division.

Response 1: The authors agree with the fact that the manuscript exhibits a great length. However, separating the article into two ones might bear the risk of losing the great benefit of a direct comparison in a single article of the most promising MC typically used for high temperature applications. The subsequent article dealing with the another monocarbide would probably yield only few scientific novelties as TaC and TiC are very similar regarding the precipitation behavior and the solubility in Co-Re which might not justify the publishing another article.

Point 2: Top of page 4: in the two-step melting process, were there no some problems of flotation of the added graphite during the second step?

Response 2: The authors did not observe significant graphite flotation in the vacuum arc melting step (second step). During the melting process, the plasma beam, figurately speaking, constantly plows through the melt and does not allow the graphite to float to the top as it is the case for melting processes there the melt stays more or less stationary, i.e. vacuum induction melting. Moreover, Co-15Re exhibits a very narrow solidification interval from approx. 1650°C to 1550°C, as seen in the binary phase diagram, which is in good accordance to our casting experience for Co-Re-Cr-TaC/TiC alloys. The rapid solidification hinders the graphite flotation to take place in larger quantities after casting.

Point 3: The Cr content is rather small (5wt.%); do you think that fast hot oxidation cannot be encountered? Are applications some protective coatings planned?

Response 3: The authors agree with you on the topic of hot oxidation issues arising from the low Cr content. Extensive research for proper oxidation resistance in Co-Re based alloys has been carried out by the research group around B. Gorr where higher content of Cr and Si in Co-Re were investigated in various studies (i.e. in Wang L, Gorr B, Christ H-J, Mukherji D, Rösler J (2013) Optimization of Cr-Content for High-Temperature Oxidation Behavior of Co–Re–Si-Base Alloys. Oxid Met. Doi:10.1007/s11085-013-9369-z). The authors decided to exclude this part of high-temperature alloy design in the introduction due to the already extensive article and focus instead on the basic requirements of the microstructures for the future assessment of the fundamental creep properties. Some experimental studies on the thermal barrier and base layer solution on Co-Re were carried out at the institute by postgraduate students but have not been published.

Point 4:

  • Line 73 : under-stochiometric®under-stoichiometric
  • Line 92: conditions:®conditions.
  • Line 166 : place Instead®place. Instead; etc … please check
  • the entire manuscript and correct the found small problems.

Response 4: The authors checked the text for the mentioned and other errors.

Point 5:  Pages 5-7: the sizes of the pictures would be decreased for the final version of the manuscript; and the arrows seem to be a little shifted and they do not show what they ought to. In fact, this is the same for the most of figures. Figure 4 must be re-arranged?

Response 4: The authors will adjust this issue before submitted the next draft.

Point 6: Figure 5: the too high proximity of the numbers of the Y-axis (…255075100…) is unpleasant.

Response 6: The font size has been reduced to so that the numbers appear less dense.

Point 7: The “Error! Reference source not found” is found several times in the text, revealing a problem of automatic linking. This is a little perturbing during reading for good understanding.

Response 7: We downloaded the submitted manuscript and checked it on my computer. It appears to be some kind of issue regarding the user’s version of Word since my document looks fine and no error statement is shown in my document. The future manuscript will be uploaded as a PDF file in order to avoid formatting issues arising from Word combability problems.

Point 8: Some whole sentences are present in some figures captions: this should be avoided

Response 8: The authors reduced caption text to an essential minimum of information which still allows the comprehension of the figures.

I look forward to hearing from and appreciate your helpful assistance.

Best regards,

Eugen Seif

Reviewer 2 Report

Dear authors,

thank you very much for sharing work on Co-Re-Cr alloys.The presented work is scientific sound and shows intersting insights on the building of carbides in this type of material. The language is adequate. The shown images are of good quality. In my opinion the research can be published.

No further comments

The referencing schould be updates

Reviewer 3 Report

After reading the article carefully,

Manuscript ID materials-2405578,

Reassessment of the matrix composition of Co-Re-Cr-based alloys for particle strengthening in high-temperature applications and investigation of suitable MC-carbides,

The article is interesting, I am interested in MC-type carbides and precipitate strengthening, but not in special purpose alloys, so I was even more interested in adding elements such as Ta, Ti, Hf. A very large structural heterogeneity was obtained, which is presented in the work, which gives it value so that it becomes useful. I was particularly interested in the obtained structure and the possibility of solution solution.

However, please correct me:

1. Text references to figures are not properly formatted, making review very difficult.

2. Please standardize the formatting of the drawings: microstructures, surface distribution of elements, XRD analyses. Text in drawings should be merged with the background.

3. Line 236, or maybe they didn't dissolve at all and stayed in a cast state?

4. Quench aging parameters are not given, which is important.

5. Where does the hydrogen in Figure 4 come from? And it's 2x C.

6. Poems 318-330, Maybe it would be possible to explain the heterogeneity of the composition in the grain areas with the conditions of crystallization?

7. Or maybe Ta and Ti will only form solid carbide precipitates, stable and not going to dissolve in the matrix, regardless of its type?

Author Response

Dear reviewer,

we kindly thank you for the review of our article and highly appreciate the time you dedicated for giving us your feedback. We are very glad that you consider our work as interesting and gave us your recommendation for the publication of this article. We thoroughly discussed your feedback and would like ask for more information on some of your questions.

Point 1: Text references to figures are not properly formatted, making review very difficult.

Response 1: We downloaded the submitted manuscript and checked it on my computer. It appears to be some kind of issue regarding the user’s version of Word since my document looks fine and no error statement is shown in my document. The future manuscript will be uploaded as a PDF file in order to avoid formatting issues arising from Word combability problems.

Point 2: Please standardize the formatting of the drawings: microstructures, surface distribution of elements, XRD analyses. Text in drawings should be merged with the background.

Response 2: The authors are currently working on a more visually pleasing and coherent design of the illustrations which will be presented later on. However, for the moment the authors would like to focus on the reviewer’s remarks regarding the content.

Point 3: Line 236, or maybe they didn't dissolve at all and stayed in a cast state?

Response 3: Due to apparent auto linking issues and prompted error messages within the text in the reviewers document, the lines of the original document and the reviewers document are mismatched. Line 236 in the original document does not mention the dissolving of particles but there is this text passage close by which might be the one you referred to: “The microstructures of Co-14.8Re-4.9Cr-1.9Ta-1.8C in Figure 3 a.) and Co-14.6Re-5.0Cr-2.1Ti-1.8C in Figure 3 c.) after ST indicate that a precipitation of fine particles has already occurred after quenching. Some large blocky TaC-particles within the matrix or at grain boundaries are also observed. Apparently, they were not dissolved during ST.” à The fact that TaC in Co-16.6Re-4.7Cr-1.5Ta-1.2C after ST (1350°C/5+1400°C/5h+1500°C/h + Ar-quenching) appears to have completely dissolved from its as-cast morphology (which is a Chinese Script morphology) as seen in Figure 12.a.) suggests that likely not all larger TaC particles in Co-14.8Re-4.9Cr-1.9Ta-1.8C just remained from the as cast state. Some may as well have formed newly during the ST treatment. Since we have no knowledge in this regard, we refrain from making any further statements.

Point 4: Quench aging parameters are not given, which is important.

Response 4: We authors checked thoroughly the article on missing heat treatment information and added these

Point 5: Where does the hydrogen in Figure 4 come from? And it's 2xC.

Response 5: We did not understand the meaning of your comment. If important to you, could you please elaborate on what you mean by hydrogen and 2xC? Might this be related to formatting issues of the auto linking problem with the figures? The element mapping results shown in Figure 4 e.) are highlighted in green and depict the distribution of Hf (Hafnium). Might there be some kind of confusion between Hf and H due to the formatting issues? This time we uploaded the draft as a PDF file in order to avoid any issues with the formation. Could you please tell us if the issues is still present in the PDF file?

Point 6: Poems 318-330, Maybe it would be possible to explain the heterogeneity of the composition in the grain areas with the conditions of crystallization?

Response 6: Due to apparent auto linking issues and prompted error messages within the text in the reviewers document, the lines of the original document and the reviewers document are mismatched. Line 318-330 in the original document does not mention any element distribution issues but there is this text passage close by which might be the one you referred to: “Figure 11 illustrates the microstructures of Co-15.3Re-4.9Cr-1.8Ta-1.8C and Co-15.1Re-5.1Cr-2.3Ti-1.8C after ST and aging at 1100°C/15h. Both microstructures do not display any Re-rich grains where no carbides precipitate but rather a homogeneous Re-distribution expressed by a uniform color distribution in the matrix. Consequently, it is assumed that no hcp-Co is present during ST. Furthermore, no lath-like structures are detected after aging at 1100°C which would indicate the presence of fcc phase during aging.” Did you refer with your question regarding the heterogeneity of the composition in the grain areas to the Re-rich grains as shown in Figure 7 and 8? If yes, could you elaborate which specific conditions of crystallization you had in mind?

Point 7: Or maybe Ta and Ti will only form solid carbide precipitates, stable and not going to dissolve in the matrix, regardless of its type?

Response 7: We know from our results that both, TaC and TiC, dissolve in the matrix during solution heat treatment and reprecipitate during cooling and aging. For example, Co-16.6Re-4.7Cr-1.5Ta-1.2C exhibits Chinese-script TaC after casting which dissolves during solution heat treatment and re-precipitates on the nanoscale after cooling and aging.

I look forward to hearing from and appreciate your helpful assistance.

Best regards,

Eugen Seif

Reviewer 4 Report

In this study, authors have reported Co-Re-Cr-based alloys for strengthening by MC-type carbides. The effect of particle-strengthening and the governing creep mechanisms of carbide-strengthened is discussed. The idea is novel and has important industrial applications.

Abstract: Novelty of the study is missing, Please highlight it.

Introduction: Please display the illustration of the study in the introduction. Similar illustration for experimental study to make it more attractive. Please highlight the novelty and purpose of the study. It is mentioned by I would suggest dividing the last paragraph into novelty and purpose of the study.

Result and discussion: Figure 5 Why XRD peaks are not clear, Please explain it. All references and cited Figures have errors please correct them. The effect of strengthening must be addressed. I would suggest adding references in the introduction section for comparable simulation results from the literature.

https://www.sciencedirect.com/science/article/pii/S1738573319300543?via%3Dihub

https://www.sciencedirect.com/science/article/pii/S2238785422016696

Author Response

Dear reviewer,

we kindly thank you for the review of our article and highly appreciate the time you dedicated for giving us your feedback. We are very glad that you consider our work as interesting and gave us your recommendation for the publication of this article. We thoroughly discussed your feedback and made according adjustments (highlighted in blue letters) which you can find in the PDF attached to this message. This time we uploaded the draft as a PDF file in order to avoid any issues with the formation.

Point 1: Abstract: Novelty of the study is missing, Please highlight it.

Response 1: The authors highlighted the novelty of the study by adjusting the wording so that the novelty becomes obvious.

Point 2: Introduction:

  1. Please display the illustration of the study in the introduction. Similar illustration for experimental study to make it more attractive.
  2. Please highlight the novelty and purpose of the study. It is mentioned by I would suggest dividing the last paragraph into novelty and purpose of the study.

Response 2:

  1. We did not understand the meaning of your comment. If important to you, could you please elaborate the meaning of the “illustration of the study in the introduction and experimental study”? Do you mean an actual depiction of the experimental study? I.e. an overview of the casting plan and methods used such as XRD, EDX and SEM?
  2. The authors divided the last paragraph into two subparagraphs corresponding to “purpose” and “novelty”. Furthermore, the wording was adjusted to highlight the intention of these subparagraphs and a new reference was added for better understanding of the novelty of TiC and HfC in Co-Re based alloys.

Point 3: Result and discussion:

  1. Figure 5 Why XRD peaks are not clear, Please explain it.
  2. All references and cited Figures have errors please correct them.
  3. The effect of strengthening must be addressed.
  4. I would suggest adding references in the introduction section for comparable simulation results from the literature.

Response 3:

  1. We assume your question refers to the weak peaks of the Hf-containing phases. Given the low Hf-content of Co-13.5Re-3.2Cr-2.3Hf-2.4C, which may be furthermore distributed between different phases (Co7Hf2, Co7Hf, and Co23Hf6 and HfC), low volume fractions of the corresponding phases result. This leads to a poor signal to background ratio and correspondingly to weak peaks which are not very distinct. We have addressed this issue by the following sentences in the article:
    1. “The weakness of the diffraction peak emphasizes the low volume fraction of HfC.”
    2. “The low volume fraction of the Co-Hf phases in comparison to the Co-Re-Cr matrix only allows the detection of their most pronounced peaks which all coincide between 44° and 45°.”
    3. “Unfortunately, the second-strongest peaks at 40.8° and 37.2°for Co23Hf6 and Co7Hf2, respectively, cannot be detected because of the comparatively low volume fraction of these phases.”

Therefore, we believe that this aspect has been sufficiently explained.

  1. We downloaded the submitted manuscript and checked it on my computer. It appears to be some kind of issue regarding the user’s version of Word since my document looks fine and no error statement is shown in my document. The future manuscript will be uploaded as a PDF file in order to avoid formatting issues arising from Word combability problems.
  2. The effect on strengthening of the presented microstructures is difficult to address since no (thermo-)mechanical experimental tests (i.e. creep tests or compression tests at room temperature) to quantitively assess the strengthening s are presented in this article. However, a general statement for the coarse Hf-phases (Co7Hf2, Co7Hf, Co23Hf6 and HfC) regarding the assumingly inferior strengthening in creep application can be added in the discussion of the section “Solution treated and aged microstructure of Co-Re-Cr-Ta-C, Co-Re-Cr-Ti-C and Co-Re-Cr-Hf-C”: “It illustrates a very low content of Hf in the matrix as the majority of Hf is bound either as coarse HfC or as large Co-Hf phases. Thereby, no or a negligible amount of Hf is present for the precipitation of fine particles during aging. In general, coarse particles will contribute less to the creep resistance due to the large interparticle spacing regardless of the bypass mechanism, be it climb, detachment or the Orowan-mechanism.” The author’s suggested addition regarding the strengthening is highlighted in bold letters.
  3. We are afraid to inform you that we prefer not to consider the recommended articles for the introduction for two reasons: Firstly, the recommended literature focusses on simulation studies whereas this article has a pure experimental approach. Secondly, the research in the suggested literature deals with material systems which show no relation to the investigated Co-Re system in this article. For these reasons, we think that those articles neither fit into the general section of the introduction where the need for non-Ni-based high temperature alloys is addressed nor into the more specific section with details on Co-Re alloys.

I look forward to hearing from and appreciate your helpful assistance.

Best regards,

Eugen Seif
